# Unsupervised Learning of Object Keypoints for Perception and Control

Tejas Kulkarni*[1], Ankush Gupta*[1], Catalin Ionescu[1], Sebastian Borgeaud[1],
Malcolm Reynolds[1], Andrew Zisserman[1,2], and Volodymyr Mnih[1]

\* indicates equal contribution

[1]*DeepMind, London*
[2]*VGG, Department of Engineering Science, University of Oxford*
{tkulkarni,ankushgupta,cdi,sborgeaud,mareynolds,zisserman,vmnih}@google.com

## Abstract

The study of object representations in computer vision has primarily focused on developing representations that are useful for image classification, object detection, or semantic segmentation as downstream tasks. In this work we aim to learn object representations that are useful for control and reinforcement learning (RL). To this end, we introduce *Transporter*, a neural network architecture for discovering concise geometric object representations in terms of *keypoints* or image-space coordinates. Our method learns from raw video frames in a fully unsupervised manner, by *transporting* learnt image features between video frames using a keypoint bottleneck. The discovered keypoints track objects and object parts across long time-horizons more accurately than recent similar methods. Furthermore, consistent long-term tracking enables two notable results in control domains – (1) using the keypoint co-ordinates and corresponding image features as inputs enables highly sample-efficient reinforcement learning; (2) learning to explore by controlling keypoint locations drastically reduces the search space, enabling deep exploration (leading to states unreachable through random action exploration) without any extrinsic rewards. Code for the model is available at: https://github.com/deepmind/deepmind-research/tree/master/transporter.

## 1 Introduction

*End-to-end learning* of feature representations has led to advances in image classification [19], generative modeling of images [8] and agents which outperform expert humans at game play [24, 31]. However, this training procedure induces task-specific representations, especially in the case of reinforcement learning, making it difficult to re-purpose the learned knowledge for future unseen tasks. On the other hand, humans explicitly learn notions of objects, relations, geometry and cardinality in a task-agnostic manner [32] and re-purpose this knowledge to future tasks. There has been extensive research inspired by psychology and cognitive science on explicitly learning object-centric representations from pixels. Both instance and semantic segmentation has been approached using supervised [23, 26] and unsupervised learning [3, 10, 15, 11, 17, 22, 7] methods. However, the representations learned by these methods do not explicitly encode fine-grained locations and orientations of object parts, and thus they have not been extensively used in the control and reinforcement learning literature. We argue that being able to precisely control objects and object parts is at the root of many complex sensory motor behaviours.

In recent work, object keypoint or landmark discovery methods [40, 16] have been proposed to learn representations that precisely represent locations of object parts. These methods predict a set of Cartesian co-ordinates of keypoints denoting the salient locations of objects given image frame(s).

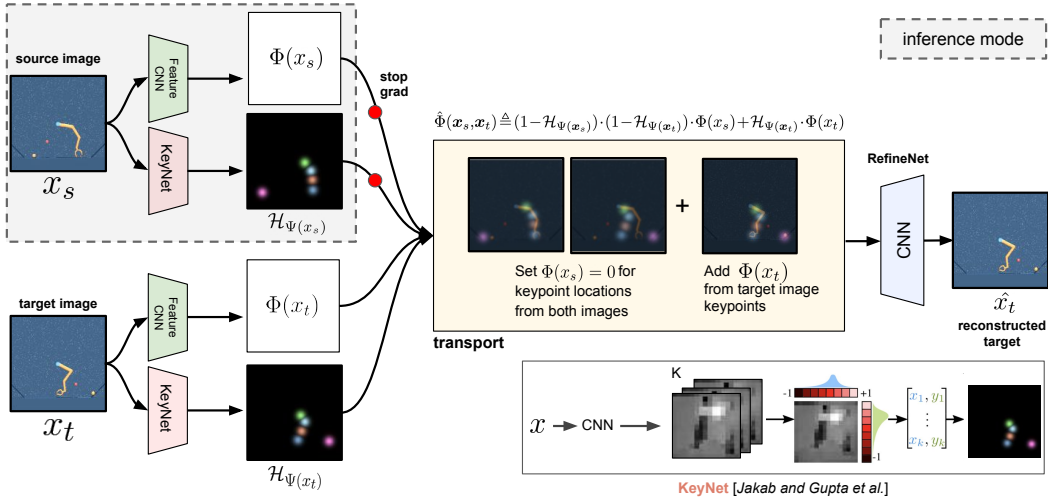

Figure 1: **Transporter.** Our model leverages object motion to discover keypoints by learning to transform a source video frame ($\boldsymbol{x}_s$) into another target frame ($\boldsymbol{x}_t$) by *transporting* image features at the discovered object locations. During training, spatial feature maps $\Phi(\boldsymbol{x})$ and keypoints co-ordinates $\Psi(\boldsymbol{x})$ are predicted for both the frames using a ConvNet and the fully-differentiable *KeyNet* [16] respectively. The keypoint co-ordinates are transformed into Gaussian heatmaps (same spatial dimensions as feature maps) $\mathcal{H}_{\Psi(\boldsymbol{x})}$. We perform two operations in the transport phase: (1) the features of the source frame are set to zero at both locations $\mathcal{H}_{\Psi(\boldsymbol{x}_s)}$ and $\mathcal{H}_{\Psi(\boldsymbol{x}_t)}$; (2) the features in the source image $\Phi(x_s)$ at the target positions $\Psi(\boldsymbol{x}_t)$ are replaced with the features from the target image $\mathcal{H}_{\Psi(\boldsymbol{x}_t)} \cdot \Phi(x_t)$. The final refinement ConvNet (which maps from the transported feature map to an image) then has two tasks: (i) to inpaint the missing features at the source position; and (ii) to clean up the image around the target positions. During inference, keypoints can be extracted for a *single* frame via a feed-forward pass through the *KeyNet* ($\Psi$).

However, as we will show, the existing methods struggle to accurately track keypoints under the variability in number, size, and motion of objects present in common RL domains.

We propose *Transporter*, a novel architecture to explicitly discover spatially, temporally and geometrically aligned keypoints given only videos. After training, each keypoint represents and tracks the co-ordinate of an object or object part even as it undergoes deformations (see fig. 1 for illustrations). As we will show, *Transporter* learns more accurate and more consistent keypoints on standard RL domains than existing methods. We will then showcase two ways in which the learned keypoints can be used for control and reinforcement learning. First, we show that using keypoints as inputs to policies instead of RGB observations leads to drastically more data efficient reinforcement learning on Atari games. Second, we show that by learning to control the Cartesian coordinates of the keypoints in the image plane we are able to learn skills or options [33] grounded in pixel observations, which is an important problem in reinforcement learning. We evaluate the learned skills by using them for exploration and show that they lead to much better exploration than primitive actions, especially on sparse reward tasks. Crucially, the learned skills are task-agnostic because they are learned without access to any rewards.

In summary, our key contributions are:

- *Transporter* learns state of the art object keypoints across a variety of commonly used RL environments. Our proposed architecture is robust to varying number, size and motion of objects.

- Using learned keypoints as state input leads to policies that perform better than state-of-the-art model-free and model-based reinforcement learning methods on several Atari environments, while using only up to 100k environment interactions.

- Learning skills to manipulate the most controllable keypoints provides an efficient action space for exploration. We demonstrate drastic reductions in the search complexity for exploring challenging Atari environments. Surprisingly, our action space enables random agents to play several Atari games without rewards and any task-dependent learning.

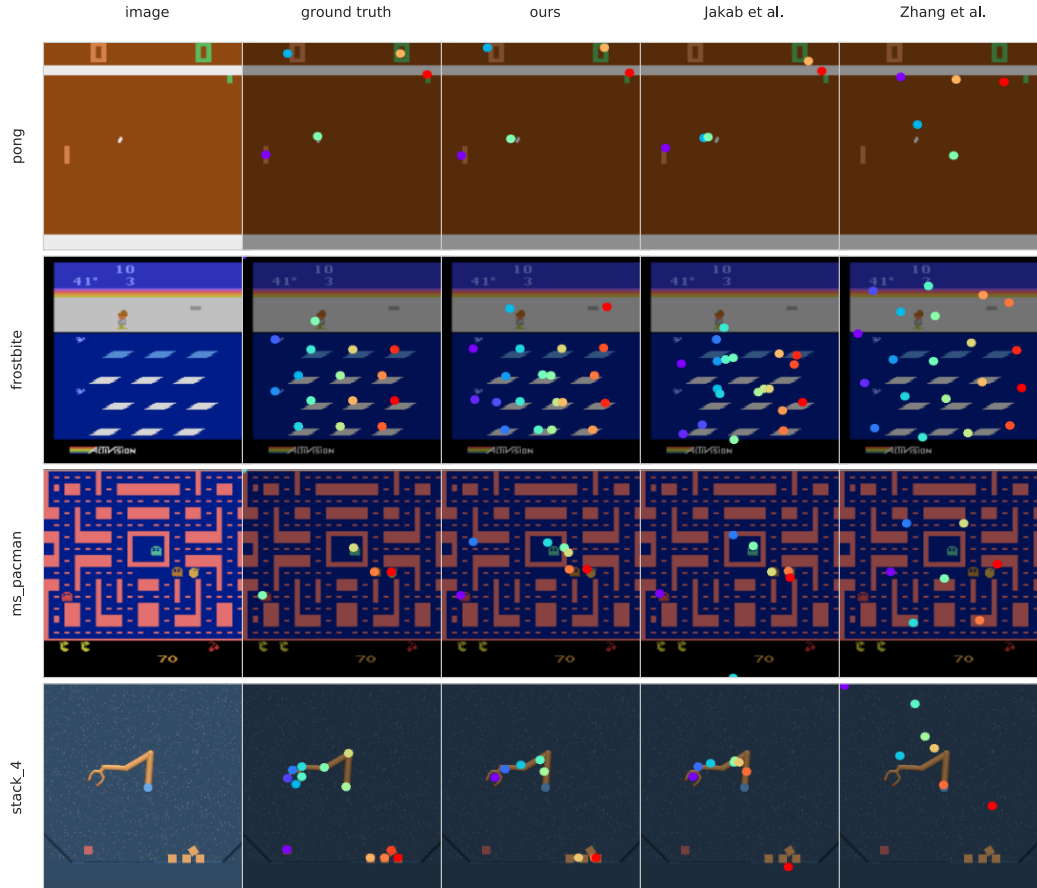

Figure 2: **Keypoint visualisation.** Visualisations from our and state-of-the-art unsupervised object keypoint discovery methods: Jakab *et al.* [16] and Zhang *et al.* [40] on Atari ALE [1] and Manipulator [35] domains. Our method learns more spatially aligned keypoints, *e.g.* `frosbite` and `stack_4` (see section 4.1). Quantitative evaluations are given in fig. 4 and further visualisations in the supplementary material.

## 2   Related Work

Our work is related to the recently proposed literature on unsupervised object keypoint discovery [40, 16]. Most notably, Jakab and Gupta et al. [16] proposed an encoder-decoder architecture with a differentiable keypoint bottleneck. We reuse their bottleneck architecture but add a crucial new inductive bias – the feature transport mechanism – to constrain the representation to be more spatially aligned compared to all baselines. The approach in Zhang et al. [40] discovers keypoints using single images and requires privileged information about temporal transformations between frames in form of optical flow. This approach also requires multiple loss and regularization terms to converge. In contrast, our approach does not require access to these transformations and learns keypoints with a simple pixel-wise L2 loss function. Other works similarly either require known transformations or output dense correspondences instead of discrete landmarks [36, 30, 34, 38]. Deep generative models with structured bottlenecks have recently seen a lot of advances [4, 20, 37, 39, 13] but they do not explicitly reason about geometry.

Unsupervised learning of object keypoints has not been widely explored in the control literature, with the notable exception of [6]. However, this model uses a full-connected layer for reconstruction and therefore can learn non-spatial latent embeddings similar to a baseline we consider [16]. Moreover, similar to [40] their auto-encoder reconstructs *single* frames and hence does not learn to factorize geometry. Object-centric representations have also been studied in the context of intrinsic motivation, hierarchical reinforcement learning and exploration. However, existing approaches either require hand-crafted object representations [21] or have not been shown to capture fine-grained representations over long temporal horizons [15].

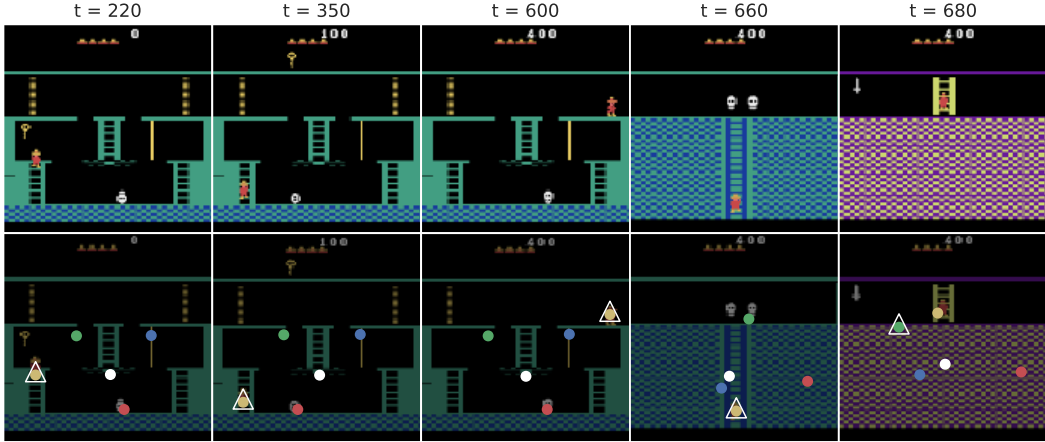

Figure 3: **Temporal consistency of keypoints.** Our learned keypoints are temporally consistent across hundreds of environment steps, as demonstrated in this classical hard exploration game called montezuma's revenge [1]. Additionally, we also predict the most controllable keypoint denoted by the triangular markers, without using any environment rewards. This prediction often corresponds to the avatar in the game and it is consistently tracked across different parts of the state space. See section 4.2.2 for further discussion.

## 3 Method

In section 3.1 we first detail our model for unsupervised discovery of object keypoints from videos. Next, we describe the application of the learned object keypoints to control for – (1) data-efficient reinforcement learning (section 3.2.1) and (2) learning keypoint based options for efficient exploration (section 3.2.2).

### 3.1 Feature Transport for learning Object Keypoints

Given an image $\boldsymbol{x}$, our objective is to extract $K$ 2-dimensional image locations or *keypoints*, $\Psi(\boldsymbol{x}) \in \mathbb{R}^{K \times 2}$, which correspond to locations of objects or object-parts without any manual labels for locations. We follow the formulation of [16] and assume access to frame pairs $\boldsymbol{x}_s, \boldsymbol{x}_t$ collected from some trajectories such that the frames differ only in objects' pose / geometry or appearance. The learning objective is to reconstruct the second frame $\boldsymbol{x}_t$ from the first $\boldsymbol{x}_s$. This is achieved by computing ConvNet (CNN) feature maps $\Phi(\boldsymbol{x}_s), \Phi(\boldsymbol{x}_t) \in \mathbb{R}^{H' \times W' \times D}$ and extracting $K$ 2D locations $\Psi(\boldsymbol{x}_s), \Psi(\boldsymbol{x}_t) \in \mathbb{R}^{K \times 2}$ by marginalising the keypoint-detetor feature-maps along the image dimensions (as proposed in [16]). A *transported* feature map $\hat{\Phi}(\boldsymbol{x}_s, \boldsymbol{x}_t)$ is generated by suppressing both sets of keypoint locations in $\Phi(\boldsymbol{x}_s)$ and compositing in the featuremaps around the keypoints from $\boldsymbol{x}_t$:

$$\hat{\Phi}(\boldsymbol{x}_s, \boldsymbol{x}_t) \triangleq (1 - \mathcal{H}_{\Psi(\boldsymbol{x}_s)}) \cdot (1 - \mathcal{H}_{\Psi(\boldsymbol{x}_t)}) \cdot \Phi(x_s) + \mathcal{H}_{\Psi(\boldsymbol{x}_t)} \cdot \Phi(x_t) \quad (1)$$

where $\mathcal{H}_\Psi$ is a heatmap image containing fixed-variance isotropic Gaussians around each of the $K$ points specified by $\Psi$. A final CNN with small-receptive field refines the transported reconstruction $\hat{\Phi}(\boldsymbol{x}_s, \boldsymbol{x}_t)$ to regress the target frame $\hat{\boldsymbol{x}}_t$. We use pixel-wise squared-$\ell_2$ reconstruction error $||\boldsymbol{x}_t - \hat{\boldsymbol{x}}_t||_2^2$ for end-to-end learning. The keypoint network $\Psi$ learns to track moving entities between frames to enable successful reconstruction.

In words, (i) the features in the source image $\Phi(x_s)$ at the target positions $\Psi(\boldsymbol{x}_t)$ are replaced with the features from the target image $\mathcal{H}_{\Psi(\boldsymbol{x}_t)} \cdot \Phi(x_t)$ — this is the *transportation*; and (ii) the features at the source position $\Psi(\boldsymbol{x}_s)$ are set to zero. The refine net (which maps from the transported feature map to an image) then has two tasks: (i) to inpaint the missing features at the source position; and (ii) to clean up the image around the target positions. Refer to fig. 1 for a concise description of our method.

Note, unlike [16] who regress the target frame from *stacked* target keypoint heatmaps $\mathcal{H}_{\Psi(\boldsymbol{x}_t)}$ and source image features $\Phi(x_s)$, we enforce explicit *spatial transport* for stronger correlation with image locations leading to more robust long-term tracking (section 4.1).

## 3.2 Object Keypoints for Control

Consider a Markov Decision Process (MDP) $(\mathcal{X}, \mathcal{A}, \mathcal{T}, r)$ with pixel observations $x \in \mathcal{X}$ as states, actions $a \in \mathcal{A}$, transition function $T : \mathcal{X} \times \mathcal{A} \to \mathcal{X}$ and reward function $r : \mathcal{X} \to \mathbb{R}$. *Transporter* keypoints provide a concise visual state-abstraction which enables faster learning (section 3.2.1). Further, in 3.2.2 we demonstrate task-indepedent exploration by learning to control the keypoint coordiantes.

### 3.2.1 Data-efficient reinforcement learning

Our first hypothesis is that task-agnostic learning of object keypoints can enable fast learning of a policy. This is because once keypoints are learnt, the control policy can be much simpler and does not have to relearn visual features using temporal difference learning. In order to test this hypothesis, we use a variant of the neural fitted Q-learning framework [28] with learned keypoints as input and a recurrent neural network Q function to output behaviors. Specifically, the agent observes the *thermometer encoded* [2] keypoint coordinates $\Psi(x_t)$, and also the features $\Phi(x_t)$ under the keypoint locations obtained by spatially averaging the feature tensor ($\Phi$) multiplied with (Gaussian) heat-maps ($\mathcal{H}_\Psi$). *Transporter* is pre-trained by collecting data using a random policy and without any reward functions (see supplementary material for details). Then, *Transporter* network weights are fixed during behavior learning from environment rewards.

### 3.2.2 Keypoint-based options for efficient exploration

Our second hypothesis is that learned keypoints can enable significantly better task-independent exploration. Typically, raw actions are randomly sampled to bootstrap goal-directed policy learning. This exploration strategy is notoriously inefficient. We leverage the Transporter representation to learn a new action space. The actions are now skills grounded in the control of co-ordinate values of each keypoint. This idea has been explored in the reinforcement learning community [21, 15] but it has been hard to learn spatial features with long temporal consistency. Here we show that *Transporter* is particularly amenable to this task. We show that randomly exploring in this space leads to significantly more rewards compared to raw actions. Our learned action space is agnostic to the control algorithm and hence other exploration algorithms [25, 5, 27] can also benefit from using it.

To do this, we define $K \times 4$ intrinsic reward functions using the keypoint locations, similar to the *VisEnt* agent [15]. Each reward function corresponds to how much each keypoint moves in the 4 cardinal directions (up, down, left, right) between consecutive observations. We learn a set of $K \times 4$ Q functions $\{Q_{i,j} | i \in \{1, ..., K\}, j \in \{1, 2, 3, 4\}\}$ to maximise each of the following reward functions: $r_{i,1} = \Psi_x^i(x_{t+1}) - \Psi_x^i(x_t)$, $r_{i,2} = \Psi_x^i(x_t) - \Psi_x^i(x_{t+1})$, $r_{i,3} = \Psi_y^i(x_{t+1}) - \Psi_y^i(x_t)$, $r_{i,4} = \Psi_y^i(x_t) - \Psi_y^i(x_{t+1})$. These functions correspond to increasing/decreasing the $x$ and $y$ coordinates respectively. The $Q$ functions are trained using n-step $Q(\lambda)$.

During training, we randomly sample a particular Q function to act with and commit to this choice for $T$ timesteps before resampling. All Q functions are trained using experiences generated from all policies via a shared replay buffer. Randomly exploring in this Q space can already reduce the search space as compared to raw actions. During evaluation, we further reduce this search space using a fixed *controllability policy* $\pi_{Q\text{gap}}$ to select the single "most controllable" keypoint, where

$$\pi_{Q\text{gap}}(s) = \underset{i}{\arg\max} \frac{1}{4} \sum_{j=1}^{4} \max_a Q_{i,j}(s; a) - \min_a Q_{i,j}(s; a). \tag{2}$$

$\pi_{Q\text{gap}}$ picks keypoints for which actions lead to more prospective change in all spatial directions than all other keypoints. For instance, in Atari games this corresponds to the avatar which is directly controllable on the screen. Our random exploration policy commits to the $Q_{ij}$ function corresponding to the keypoint $i$ selected as above and a direction $j$ sampled uniformly at random for $T$ timesteps and then resamples. Consider a sequence of 100 actions with 18 choices before receiving rewards, which is typical in hard exploration Atari games (e.g. montezuma's revenge). A random action agent would need to search in the space of $18^{100}$ raw actions. However, observing 5 keypoints and $T = 20$ only has $(5 \times 4)^{100/20}$, giving a search space reduction of $10^{100}$. The search space reduces further when we explore with the most controllable keypoints. Since our learned action space is agnostic to the control mechanism, we evaluate them by randomly searching in this space versus raw actions. We measure extrinsically defined game score as the metric to evaluate the effectiveness of both search procedures.

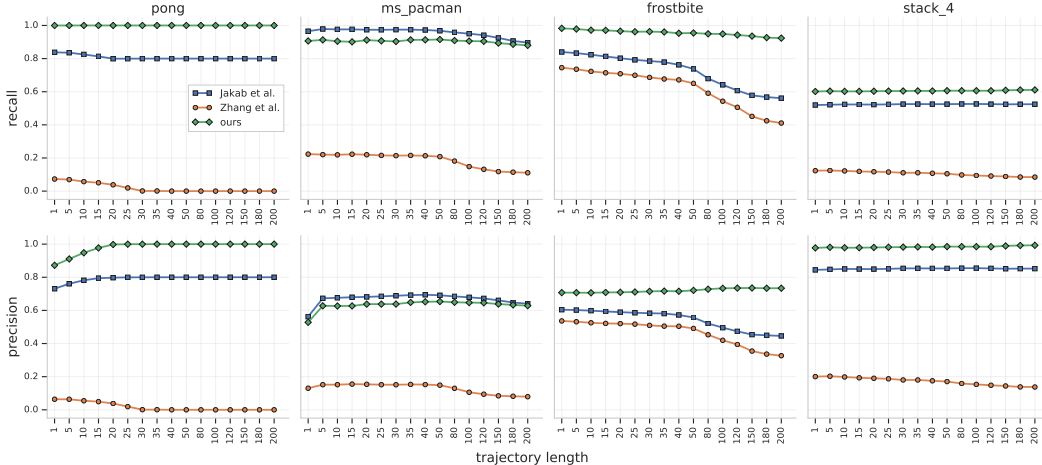

Figure 4: **Long-term tracking evaluation.** We compare long-term tracking ability of our keypoint detector against Jakab *et al.* [16] and Zhang *et al.* [40] (visualisations in fig. 2 and supplementary material). We report precision and recall for trajectories of varying lengths (lengths $= 1 - 200$ frames; each frame corresponds to 4 action repeats) against ground-truth keypoints on Atari ALE [1] and Manipulator [35] domains. Our method significantly outperforms the baselines on all games ($100\%$ on pong), except for ms_pacman where we perform similarly especially for long trajectories (length $= 200$). See section 4.1 for further discussion.

## 4 Experiments

In section 4.1 we first evaluate the long-term tracking ability of our object keypoint detector. Next, in section 4.2 we evaluate the application of the keypoint detector on two control tasks — comparison against state-of-the-art model-based and model-free methods for data-efficient learning on Atari ALE games [1] in section 4.2.1, and in section 4.2.2 examine efficient exploration by learning to control the discovered keypoints; we demonstrate reaching states otherwise unreachable through random explorations on raw-actions, and also recover the agent *self* as the most-controllable keypoint. For implementation details, please refer to the supplementary material.

**Datasets.** We evaluate our method on Atari ALE [1] and Manipulator [35] domains. We chose representative levels with large variations in the type and number of objects. (1) For evaluating long-term tracking of object keypoints section 4.1 we use — pong, frostbite, ms_pacman, and stack_4 (manipulator with blocks). (2) For data-efficient reinforcement learning (section 4.2.1) we train on diverse data collected using random exploration on the Atari games indicated in fig. 6. (3) For keypoints based efficient-exploration (section 4.2.2) we evaluate on one of the most difficult exploration game — montezuma revenge, along with ms_pacman and seaquest.

A random policy executes actions and we collect a trajectory of images before the environment resets; details for data generation are presented in the supplementary material. We sample the source and target frames $\boldsymbol{x}_s, \boldsymbol{x}_t$ randomly within a temporal offset of 1 to 20 frames, corresponding to small or significant changes in the the configuration between these two frames respectively. For Atari ground-truth object locations are extracted from the emulated RAM using hand crafted per-game rules and for Manipulator it is extracted from the simulator geoms. The number of keypoints $K$ is set to the maximum number of moving entities in each environment.

### 4.1 Evaluating Object Keypoint Predictions

**Baselines.** We compare our method against state-of-the-art methods for unsupervised discovery of object landmarks — (1) Jakab *et al.* [16] and (2) Zhang *et al.* [40]. For (1) we use exactly the same architecture for $\Phi$ and $\Psi$ as ours; for (2) we use the implementation released online by the authors where the image-size is set to $80 \times 80$ pixels. We train all the methods for $10^6$ optimization steps and pick the best model checkpoint based on a validation set.

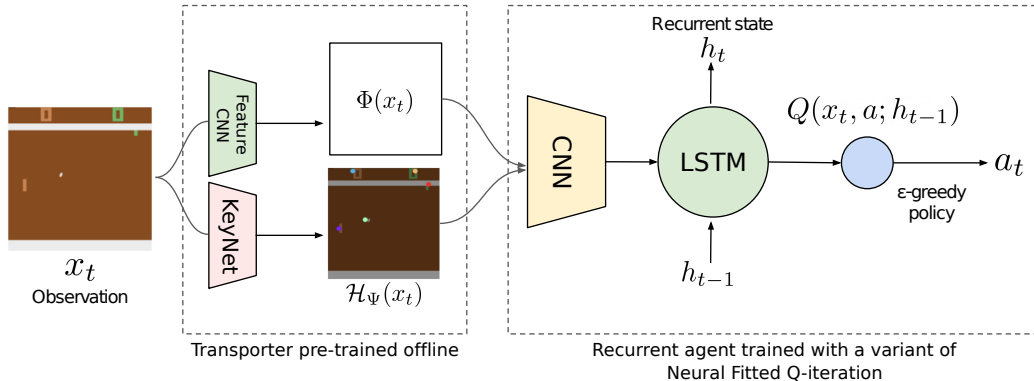

Figure 5: **Agent architecture for data-efficient reinforcement learning.** *Transporter* is trained off-line with data collected using a random policy. A recurrent variant of the neural-fitted Q-learning algorithm [28] rapidly learns control policies using keypoint co-ordinates and features at the corresponding locations given game rewards.

| Game | KeyQN (ours) | SimPLe | Rainbow | PPO (100k) | Human | Random |
|---|---|---|---|---|---|---|
| breakout | 19.3 (4.5) | 12.7 (3.8) | 3.3 (0.1) | 5.9 (3.3) | 31.8 | 1.7 |
| frostbite | 388.3 (142.1) | 254.7 (4.9) | 140.1 (2.7) | 174.0 (40.7) | 4334.7 | 65.2 |
| ms_pacman | 999.4 (145.4) | 762.8 (331.5) | 364.3 (20.4) | 496.0 (379.8) | 15693.0 | 307.3 |
| pong | 10.8 (5.7) | 5.2 (9.7) | -19.5 (0.2) | -20.5 (0.6) | 9.3 | -20.7 |
| seaquest | 236.7 (22.2) | 370.9 (128.2) | 206.3 (17.1) | 370.0 (103.3) | 20182.0 | -20.7 |

Figure 6: **Atari Mean Scores**. Mean scores (and std-dev in parentheses) obtained by our method (three random seeds) in comparison with Rainbow [12], SimPLe [18] and PPO [29] trained on 100K steps (400K frames). See section 4.2.1 for details. Numbers (except for KeyQN) taken from [18].

**Metrics.** We measure the precision and recall of the detected keypoint trajectories, varying their lengths from 1 to 200 frames (200 frames $\approx$ 13 seconds @ 15-fps with action-repeat of 4) to evaluate long-term consistency of the keypoint detections crucial for control. The average Euclidean distance between each detected and ground-truth trajectory is computed. The time-steps where a ground-truth object is absent are ignored in the distance computation. Distances above a threshold ($\epsilon$) are excluded as potential matches.[1] One-to-one assignments between the trajectories are then computed using min-cost linear sum assignment, and the matches are used for reporting precision and recall.

**Results.** Figure 2 visualises the detections while fig. 4 presents precision and recall for varying trajectory lengths. *Transporter* consistently tracks the salient object keypoints over long time horizons and outperforms the baseline methods on all environments, with the notable exception of [16] on `pacman` where our method is slightly worse but achieves similar performance for long-trajectories.

## 4.2 Using Keypoints for Control

### 4.2.1 Data-efficient Reinforcement Learning on Atari

We demonstrate that using the learned keypoints and corresponding features within a reinforcement learning context can lead to data-efficient learning in Atari games. Following [18], we trained our Keypoint Q-Network (KeyQN) architecture for 100,000 interactions, which corresponds to 400,000 frames. As shown in fig. 6, our approach is better than the state-of-the-art model-based SimPLe architecture [18], as well as the model-free Rainbow architecture [12] on four out of five games. Applying

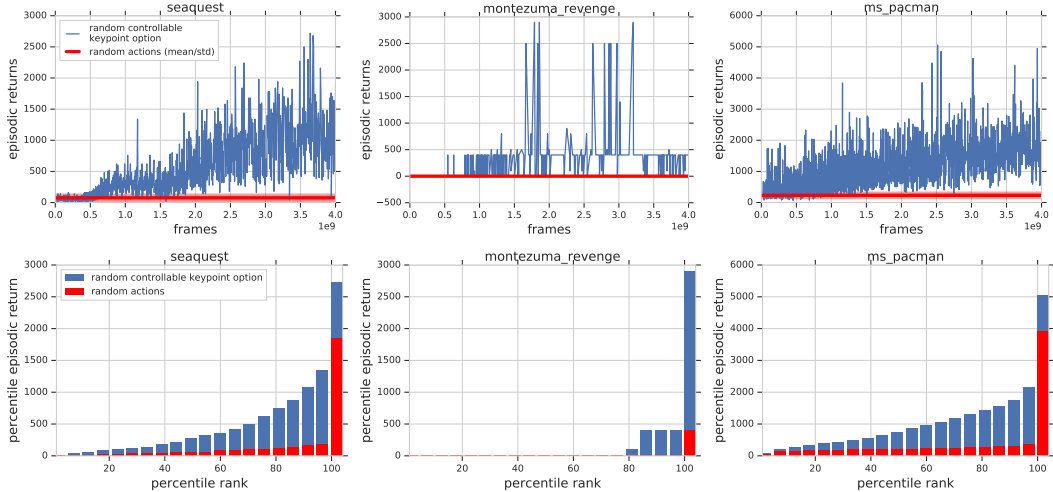

Figure 7: **Exploration using random actions versus random (most controllable) keypoint option / skills**: *(first row)* We perform random actions in the environment for all methods (without reward) and record the mean and standard deviation of episodic returns across 4 billion frames. With the same frame budget, we simultaneously learn the most controllable keypoint and randomly explore in the space of its co-ordinates (to move it *left, right, top, down*). The options model becomes better with training (using only intrinsic rewards) and this leads to higher extrinsically defined episodic returns. Surprisingly, our learned options model is able to play several Atari games via random sampling of options. This is possible by learning skills to move the discovered game avatar as far as possible without dying. *(second row)* We measure the percentile episodic return reached for all methods. Our approach outperforms the baseline, both in terms of efficient and robust exploration of rare and rewarding states.

this approach to all Atari games will require training *Transporter* inside the reinforcement learning loop because pre-training keypoints on data from a random policy is insufficient for games where new objects or screens can appear. However, these experiments provide evidence that the right visual abstractions and simple control algorithms can produce highly data efficient reinforcement learning algorithms.

### 4.2.2 Efficient Exploration with Keypoints

We learn options/skills for efficient exploration from the object keypoints. We use a distributed off-policy learner similar to [14] using 128 actors and 4 GPUs. The agent network has a standard CNN architecture [24] with an LSTM with 256 hidden units which feeds into a linear layer with $K \times 4 \times a$ units, where $a$ is the number of actions. Our *Transporter* model is learnt simultaneously with all the control policies (no pre-training). We commit to the choice of Q function (corresponding to a keypoint and one of the four directions) for $T = 20$ steps (see section 3.2.2 for details). Actions are sampled using an $\epsilon$-greedy random policy during training ($\epsilon$ is sampled from a log-uniform distribution over [1e-4, 0.4]), and greedily for evaluation. Quantitative results are shown in fig. 7. We also show qualitative results of the most controllable keypoint in fig. 3 and the supplementary material.

Our experiments clearly validate our hypothesis that using keypoints enables temporally extended exploration. As shown in fig. 7, our learned keypoint options consistently outperform the random actions baseline by a large margin. Encouragingly, our random options policy is able to play some Atari games by moving around the avatar (most controllable keypoint) in different parts of the state space without dying. For instance, the agent explores multiple rooms in Montezuma's Revenge, a classical hard exploration environment in the reinforcement learning community. Similarly, our keypoint exploration learns to consistently move around the submarine in Seaquest and the avatar in Ms. Pacman. Most notably, this is achieved without rewards or (extrinsic) task-directed learning. Therefore our learned keypoints are stable enough to learn complex object-oriented skills in the Atari domain.

# 5   Conclusion

We demonstrate that it is possible to learn stable object keypoints across thousands of environment steps, without having access to task-specific reward functions. Therefore, object keypoints could provide a flexible and re-purposable representation for efficient control and reinforcement learning. Scaling keypoints to work reliably on richer datasets and environments is an important future area of research. Further, tracking objects over long temporal sequences can enable learning object dynamics and affordances which could be used to inform learning policies. A limitation of our model is that we do not currently handle moving backgrounds. Recent work [9] that explicitly reasons about camera / ego motion could be integrated to globally transport features between source and target frames. In summary, our experiments provide clear evidence that it is possible to learn visual abstractions and use simple algorithms to produce highly data efficient control policies and exploration procedures.

**Acknowledgements.** We thank Loic Matthey and Relja Arandjelović for valuable discussions and comments.

## Footnotes

[1]The threshold value ($\epsilon$) for evaluation is set to the average ground-truth spatial extent of entities for each environment (see supplementary material for details).

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
