[Supplementary Material]

# Unsupervised Learning of Object Keypoints for Perception and Control
## Supplementary Material

Tejas Kulkarni*[1], Ankush Gupta*[1], Catalin Ionescu[1], Sebastian Borgeaud[1],
Malcolm Reynolds[1], Andrew Zisserman[1,2], and Volodymyr Mnih[1]

\* indicates equal contribution

[1]*DeepMind, London*
[2]*VGG, Department of Engineering Science, University of Oxford*
{tkulkarni,ankushgupta,cdi,sborgeaud,mareynolds,zisserman,vmnih}@google.com

## 1 Implementation Details

The feature extractor $\Phi$ is a convolutional neural network with six `Conv-BatchNorm-ReLu` layers [2]. The filter size for the first layer is $7{\times}7$ with 32 filters, and $3{\times}3$ for the rest with number of filters doubled after every two layers. The stride was set to 2 for layer 3 and 5 (1 for the rest). *KeyNet* $\Psi$ has a similar architecture but includes a final $1{\times}1$ regressor to $K$ feature-maps corresponding to $K$ keypoints. 2D co-ordinates are extracted from these $K$ maps as described in [3]. The architecture of *RefineNet* is the transpose of $\Phi$ with $2\times$ bilinear-upsampling to undo striding. We specify $K$ for each environment but keep all other hyper-parameters of the network fixed across experiments. We used the Adam optimizer [4] with a learning rate of $0.001$ (decayed by $0.95$ every $10^5$ steps) and batch size of 64 across all experiments.

For evaluating keypoint predictions, the distance threshold value ($\epsilon$) was set to the average ground-truth spatial extent of entities for each environment given in the table below. Note, these $\epsilon$ values correspond to coordinates normalised to the [-1,1] range for both the $(x,y)$ dimensions.

| environment | pong | ms_pacman | frostbite | stack_4 |
|---|---|---|---|---|
| $\epsilon$ | 0.20 | 0.15 | 0.20 | 0.15 |

Code for the *Transporter* model is available at:
https://github.com/deepmind/deepmind-research/tree/master/transporter.

## 2 Diverse Data Generation

To train the Transporter, a dataset of observation pairs is constructed from environment trajectories. It is important that this dataset contains a diverse range of situations, and unconditionally storing a pair from all trajectories generated by a random policy may contain many similar pairs. To mitigate this, we use a *diverse* data generation procedure as follows.

We generate trajectories of up to length 100 (action repeat is set to 4, so these trajectories represent up to 400 environment frames) using a uniform random policy, and uniformly sample one observation from the first half of the trajectory and one frame from the second half. Trajectories are shorter than 100 only when the end of an episode is reached. A buffer containing the maximum number of pairs we want to generate (in these experiments, 100k) is populated unconditionally from a number of

environment actor threads until it is full. More frame pairs are generated, up to some defined maximum budget, and are conditionally written into the buffer as follows.

First some number of indices of existing pairs are sampled from the buffer, and for each of them we compute the nearest neighbor by L2 distance to other elements of the buffer. We take the same number of new generated frame pairs, and also compute their nearest neighbor in the buffer. For corresponding pairs of (existing frame pair, new frame pair) we overwrite the existing pair with the new pair whenever the new pair has a greater nearest neighbor distance, or if a uniform random number $\in [0,1] < 0.05$. We continue this procedure until the maximum budget is reached, and write out the final buffer as our training set. Note that the reward function is not used at all in this procedure.

For efficiency, we store a lower resolution copy of the buffer (64x64, grayscale) on the GPU to perform efficient nearest neighbor calculations, keeping corresponding higher res (128x128 RGB) copies on CPU RAM. Using a single consumer GPU and a 56 core desktop machine, with many actor subprocesses, this approach can perform 10 million environment steps (40 million total frames) in approximately 1 hour.

# 3   Videos

Videos visualising various aspects of the model are available at:
https://www.youtube.com/playlist?list=PL3LT3tVQRpbvGt5fgp_bKGvW23jF11Vi2

# 4 Pixel Transport versus Feature Transport

We investigated whether learning features is as important as spatially transporting them between frames. As shown in fig. 1, we show that transporting learned features significantly outperforms transporting pixels. Transporting pixels gives rise to ambiguous intermediate pixel representations, making it difficult for the final CNN decoder network to solve the downstream pixel prediction task. On the other hand, the feature encode higher level information and the decoder network learns a more abstract function to solve the prediction problem.

Figure 1: **Transporting features is significantly better than transporting pixels.** Given a sawyer arm with tabletop toys, *Transporter* discovers keypoints at the joint locations of the robot and object centroids (left two columns). In this experiment we investigate whether it is important to transport learned features or pixels. In case of pixel transport, the refinement network has to perform difficult and ambiguous computations to predict the target frame. Therefore the final pixel reconstruction error is significantly higher for pixel transporter (right most column).

# 5 Temporal consistency of keypoints

Figures 2, 3, 4, and 5 show the inferred keypoints on frames selected from a single episode on Atari ALE [1] (Pong, Frostbite and Ms. Pac-Man) and Manipulator [5] (stack_4) domains. The selected frames are each 10 time steps apart. The first frame has been explicitly chosen to ensure there is enough diversity in the shown frames. The colours are time consistent – a specific colour corresponds to the same keypoint throughout the episode. Thus, if a given game 'object' is assigned the same coloured keypoint throughout the episode, that keypoint is temporally consistent for that 'object'.

Videos showing the inferred keypoints by the three methods for entire episodes can be accessed at: https://www.youtube.com/playlist?list=PL3LT3tVQRpbvGt5fgp_bKGvW23jF11Vi2

Figure 2: Atari [1]: pong

Figure 3: Atari [1]: frostbite

Figure 4: Atari [1]: ms pacman

Figure 5: Manipulator [5]: stacker with 4 objects

# 6 Reconstructions

We visualise the reconstructed images on Atari ALE [1] (Figures 6, 7, and 8) and Manipulator [5] domains (Figure 9) for randomly selected frames. The rows in the figures correspond respectively to our model, Jakab and Gupta et al. [3] and Zhang et al. [6]. The first two columns are the inputs given to the models. Whereas our model requires a pair of input frames (*image* and *future_image*), the remaining two models only require single frame (*future_image*). The third column (*reconstruction*) shows the reconstructed target image. The final column (*keypoints*) shows the inferred keypoints for the given inputs.

Figure 6: Reconstruction: pong

Figure 7: Reconstruction: frostbite

Figure 8: Reconstruction: ms pacman

Figure 9: Reconstruction: stacker with 4 objects