[Reviews · NeurIPS 2019]

Reviewer 1



Response read. I appreciate the commitment to making the paper clearer. Thank you for that. This is a really good paper! ___________________________ **Summary** They learn in an unsupervised way keypoints (x,y coordinates) corresponding to “relevant” objects using the novel “Transporter” architecture, which combines an existing architecture, PointNet, with a novel unsupervised task, “feature transport” (predict the future in pixel-space, but constrain only the features at predicted keypoints to change between two consecutive frames’ feature maps). They also show how accurate their keypoint prediction is with an object tracking task using ground truth coordinates. Lastly, they use these keypoints effectively in two downstream RL tasks: model-free RL (neural fitted q iteration) with keypoint-indexed features as input and sample-efficient exploration by defining an intrinsic reward based on maximizing each keypoints movement in +x,-x,+y,-y directions. (also they find the most controllable keypoint and exploit that at test time!). They use a few games from Atari for each task. **Strengths** * The transport mechanism takes a minute to get straight, but it makes sense. It seems to be a good inductive bias that makes the pixel-based reconstruction training signal more amenable to learning good object-centric features. * The transport mechanism is also novel and seems intuitively better than the original PointNet approach of just concatenating the heat maps to the feature maps and this is shown as they track the true keypoints better than previous approaches * Also they are the first (to my knowledge) to take this keypoint based work (Jakab, Zhang) and apply to RL and show it works and helps for control and for exploration. * Their new intrinsic rewards for exploration in this new keypoint space make sense and they provably show they are better than random policy * The methods section, introduction, related work are well-written and nicely motivate and explain the transporter architecture. The paper is well organized and the diagrams nice and clear and very helpful * intrinsic reward using keypoints is novel as far I know and intuitive and promising * Results are very important/significant: learning more object-centric, state representations and using those to be more sample efficient and effective is something the field should move toward and I hope more people follow up on * Addresses exploration in a way I find intuitively better/scalable -> explicitly trying to find objects/keypoints and than operating in the space that moves them around seems like a better approach than previous work that uses pixel-space based losses to make an exploration bonus (pseudo-counts,) or surprisal-based predictive exploration (intrinsic curiosity) or exploits access to simulator (Go-Explore) -> would be cool to see the number comparisons for this though! **Weaknesses: ** * Clarity. There are parts of this paper that are a bit unclear. The diagram and caption for KeyQN section are very helpful, but the actual text section could be fleshed out more. It would nice if the text could have a little more detail on how the outputs from the transporter are input to the KeyQN architecture and how the whole thing is trained. The exploration section was well explained for most part, but it took a bit of time to understand. Maybe would help to have an algorithm box. Also, the explanation of training process a bit confusing. Maybe a diagram of the architecture and how the transporter feeds into this would help. Also, I am confused a bit about whether the transporter is pretrained and frozen or fine-tuned. One quote from the paper in this regard confused me: “Our transporter model and all control policies simultaneuosly “ so the weights of the Transporter network are not frozen during the downstream task like in KeyQN? * Experiments: They only show these results on a few games (and no error bars), so it would have been nice (but not a dealbreaker) to see results from more Atari games. They do partially justify this by saying they couldn’t use a random policy on other games, but I’d be curious just to see what happens when they try a couple more games. Would be nice to see comparisons to other exploration methods (they only show results compared to random exploration) Nitpicks/Questions * Makes sense to just refer the reader to the PointNet paper instead of re-explaining it, but a short explanation if possible of PointNet (couple sentences) might be helpful, so that one doesn’t have to skim that paper to understand this paper * The diagram in figure 5 (h_psi) should show a heat map not keypoints superimposed on raw frame right? * In the appendix “K is handpicked for each game?” How? Validation loss? * The tracking experiments but the section is a bit unclear. I have a few questions on that front: * why is there a need to separating precision and recall? * why not just report overall mean average precision or F1 score? Might be a bit easier for reader to digest one number * Why bucket into diff sequence lengths? what do the different sequence lengths mean? There is no prediction-in-keypoint space model right? So there is no concept of the performance worsening as the trajectory gets longer. Aren’t the keypoint guesses just the output of the PointNet at each frame, so why would the results from a 200 frame sequence be much different than 100 or something? Why not just report overall precision and recall on the test set? * In the KeyQN section What is the keypoint mask averaged feature vector? just multiply each feature map element wise by H_psi?

Reviewer 2



Strength: + The contributions are clearly stated. + Most of the paper is well written -- see weaknesses for exceptions. + Experiments are convincing. + To the best of my knowledge, the idea of using unsupervised keypoints for reinforcement learning is novel and promising. One can expect a variety of follow-up work. + Using keypoints as input state of a Q function is reasonable and reduces the dimensionality of the problem. + Reducing the search space to the most controllable keypoints instead of raw actions is a promising idea. Weaknesses: 1. Overstated claim on generalization In the introduction (L17-L22), the authors motivate their work by explaining that reinforcement learning approaches are limited because it is difficult to re-purpose task-specific representations, but that this is precisely what humans do. From this, one could have expected this paper to address this issue by training and applying the detector network across multiple games, re-purposing their keypoint detector. This would have be useful to verify that the learnt representations generalize to new contexts. But unfortunately, it hasn't been done, so it is a bit of an over-statement. Could this be a limitation of the method because the number of keypoints is fixed? 2. Deep RL that matters Experiments should be run multiple times. A longstanding issue with deep RL is their reproducibility and the significance of their improvements. It has been recently suggested that we need a community effort towards reproducibility [a], which should also be taken into account in this paper. Among the considerations, one critical thing is running multiple experiments and reporting the statistics. [a] Henderson, Peter, et al. "Deep reinforcement learning that matters." Thirty-Second AAAI Conference on Artificial Intelligence. 2018. 3. The choice of testing environment is not well motivated. Levels are selected without a clear rationale, with only a vague motivation in L167. This makes me suspect that they might be cherry picks. Authors should provide a more clear justification. This could be related to the next weakness that I will discuss, which is understandable. Even if this is the case, this should then be explicit with experimental evidence. 4. Keypoints are limited to moving objects A practical limitation comes from the fact that the keypoints are learnt from the moving parts of the image. As identified by the authors, the first resulting limitation is that the method assumes a fixed background, so that only meaningful objects move and can be detected as keypoints. Learning to detect keypoints based on what objects are moving has some limitations when these keypoints are supposed to be used as the input state of a Q function. One can imagine a game where some obstacles are immobile. The locations of these obstacles are important in order to make decisions but in this work, they would be ignored. It is therefore important that these limitations are also explicitly demonstrated. 5. Dealing with multiple instances. Because "PNet" generates one heatmap per keypoint, each keypoint detector "specializes" into a certain type of keypoint. This is fine for some applications (e.g. face keypoints) where only one instance of each kind of keypoint exists in each image. But there are games (e.g. Frostbite) where a lot of keypoints look exactly the same. And still, the detector is able to track them with consistency (as shown in the supplementary video). This is intriguing, as one could expect the detector to detect several keypoints at the same location, instead of distributing them almost perfectly. Is it because the receptive field is large? 6. Other issues - In section 3, the authors could improve the explanation of why the loss promotes the detection of meaningful keypoints. It is not obvious at first why the detector needs to detect keypoints to help with the reconstruction. - Figure 1: Referring to [15] as "PointNet" is confusing when this name doesn't appear anywhere in this paper ([15]) and there exists another paper with this name. See "PointNet: Deep Learning on Point Sets for 3D Classification and Segmentation", Charles R. Qi, Hao Su, Kaichun Mo, Leonidas J. Guibas. - Figure 1: The figure describes two "stop grad", but there is no mention or explanation of it in the text or caption. This is not theoretically motivated either, because most of the transported feature map comes from the source image (all the pixels that are not close from source or target keypoints). Blocking these gradients would block most of the gradients that can be used to train the "Feature CNN" and "PNet". - L93: "by marginalising the keypoint-detetor feature-maps along the image dimensions (as proposed in [15])". This would be better explained and self-contained by saying that a soft-argmax is used. - L189: "Distances above a threshold (ε) are excluded as potential matches". What threshold value is used? - What specific augmentation techniques are used during the training of the detector? - Figure 4: it is not clear what the meaning of "1-200 frames" is and how the values are computed. Why are the precision and recall changing with the trajectory length? Also, what is an "action repeat"? - Figure 6: the scores should be normalized (and maybe displayed as a plot) for easier comparison. ==== POST REBUTTAL ==== The rebuttal is quite convincing, and have addressed my concerns. I would like to raise the rating of the paper to 8 :-) I'm happy that my worries were just worries.

Reviewer 3



1. The authors propose an unsupervised representation learning task that - a) first predicts a fixed number of K keypoints (unsupervised); b) takes two frames and erases these keypoints from the features of frame A. copies the keypoint features from frame B into frame A. constructs a final feature map consisting of the modified features of frame A and keypoint features of frame B. c) a small CNN uses the feature map as input to reconstruct frame B in pixel space (l2 error). This task is interesting as the representation itself is learnt via the "feature swapping" trick guided by keypoints. 2. The experiments in the paper are geared to show that the design decisions taken by the authors are empirically valid - either in terms of how long/accurate the keypoint tracks are, or how it helps in sample efficient RL. These are two solid directions but neither answer why this task should work at all. For example, in unsupervised or self-supervised learning, most tasks have a "trivial" solution where the network learns low-level information not useful for end tasks. It is not clear what (if) the trivial solutions of this method are? The feature swapping trick must introduce discontinuities which might lead to trivial solutions? 3. Analyzing the effect of the bottleneck: How is the number of keypoints K determined and how does that effect the final performance? Is it better to set a slightly larger K in the hope that the model finds multiple useful keypoints? I think one of the simpler baselines to compare against is a denoising auto-encoder which does not have such a keypoint bottleneck but still uses a similar objective (l2 reconstruction error) to learn the representation. This will show why the keypoint bottleneck is necessary and what properties it brings. 4. Comparisons with other unsupervised feature learning methods: Methods such as [24] use a forward prediction error (curiosity) objective to learn a representation. This has been shown to be a powerful intrinsic reward. The authors should discuss their own method's empirical performance as well as theoretical comparisons. 5. Analyzing the effect of the feature representation better: I like the last experiment (Figure 7) that looks at the most controllable keypoint. I think the paper can be made stronger by considering a setting where the features themselves are kept fixed while the agent (Figure 5: Recurrent Agent) is learned. 6. Please also aim to release your code and implementation. RL algorithms are hard to reproduce.

[Author Response · NeurIPS 2019]

We thank all the reviewers for their valuable feedback.

**R1: Clarity.**   Thank you for providing detailed feedback; we will give more details as suggested. *Transporter frozen*
*for KeyQN?* Yes, Transporter is trained with frames from a random policy and frozen during policy learning (fig. 5);
this should be relaxed in the future (see also line 48).

**R1: Error bars / multiple runs. R2: Deep RL that**
**matters.**   Thank you for highlighting this.   In fig. 6 we
report the mean score of *3 runs* with different random seeds.
As suggested, here we include the standard deviations (in
parenthesis in the table on right).

| Game | KeyQN | SimPLe | Rainbow | PPO (100k) |
|---|---|---|---|---|
| breakout | 19.3  (4.5) | 12.7  (3.8) | 3.3  (0.1) | 5.9  (3.3) |
| frostbite | 388.3  (142.1) | 254.7  (4.9) | 140.1  (2.7) | 174.0  (40.7) |
| ms_pacman | 999.4  (145.4) | 762.8  (331.5) | 364.3  (20.4) | 496.0  (379.8) |
| pong | 10.8  (5.7) | 5.2  (9.7) | -19.5  (0.2) | -20.5  (0.6) |
| seaquest | 236.7  (22.2) | 370.9  (128.2) | 206.3  (17.1) | 370.0  (103.3) |

**R1, R4: Compare w/ exploration methods.**   We use the
keypoints for *options* based exploration instead of raw actions. This can be thought of as a learned action space, which
is complimentary to learning based exploration methods [26], and combinations of these can be interesting future work.

**R1, R4: How is 'K' (number of keypoints) chosen?**   We will make explicit that $K$ is a hyper-parameter. It is set to
the maximum number of moving entities in each environment. If $K$ > entities, the extra keypoints stay in a constant
position as seen in montezuma_revenge visualisation in the supplementary material.

**R1, R2: Tracking experiments.**   We will include F1-scores for brevity. *Why evaluate over different time lengths?*
Although the predictions are per-frame, this evaluation measures the consistency of correspondence of keypoints to
entities over varying time lengths, which is essential for control. It is difficult to sustain this correspondence for long
durations (*e.g.*, due to switching identities), as indicated by the general downward trend of baseline methods in fig. 4.

**R1: What is averaged feature vector in KeyQN?**   The (Gaussian) heat-map ($\mathcal{H}_\Psi$) is multiplied with the feature
tensor ($\Phi$) and then spatially averaged to obtained this feature vector. We will make this clearer.

**R2: Generalization claims.**   We demonstrate learning representations for entities (keypoints) in a task-agnostic
manner (without any rewards), which can later be re-purposed for (or generalize to) efficient task-specific reward based
learning. It is true, that a single Transporter model has not been shown to generalize across a number of environments.
This is an important form of generalization and a good candidate for future work.

**R2: Choice of testing environments + limited to moving objects.**   We did not cherry pick environments given the
empirical results and report everything we ran. We chose diverse environments that represent varying degrees of
difficulty (number and motion of entities) for the Transporter network.   For example, our model is not designed to
capture – (1) static elements like walls in ms_pacman, or (2) moving background. In environments which pose such
challenges our model is unable to capture all the relevant visual entities and performs worse than baselines, *e.g.*, bricks
in breakout where PPO-500k is better (fig. 6 in paper). We will make this discussion more explicit.

**R2: Tracking multiple instances.**   The reconstruction objective encourages detecting all the moving entities. Yes,
large receptive field captures the unique context around each object to disambiguate b/w visually similar objects.

**R2: Explain "PointNet" better.**   Thank you for the detailed comments, we will make these clearer (and rename
*PointNet*). The stop-gradient is to prevent any cross-talk / co-adaptation between the source and target images, and use
the network in inference mode for the source image. We will make this explicit and give empirical evidence.

**R2: Other comments.**   *(1) Distance threshold value ($\epsilon$).* This threshold value for evaluation was set to the average
ground-truth spatial extent of entities for each environment. We will add these details in the appendix. *(2) Augmentation*
*techniques.* The time delay between source and target frame was randomly chosen between and 1 and 20 (sec. 4); no
other augmentation strategy was used. *(3) Action repeat.* Each action was repeated 4 times (following [25]).

**R4: Trivial solutions.**   In order to reconstruct the target image from source, the network has to detect everything that
can change between two frames, *i.e.* learn to detect the moving entities. In Atari ALE, often objects of interest move,
hence the proposed approach provides useful geometric representations. For limitations see line 28 above.

**R4: Why is keypoint bottleneck necessary?**   Our objective is to learn object representations that are geometric and
correspond to discrete entities. Such representations enable learning co-ordinate based options models for exploration
and efficient reward based control. Moreover, in previous work [16], generic fully-connected representations of an
auto-encoder were shown to be deficient for learning object keypoints (ref: ablation in table 2 of [16]).

**R4: Fix representation, learn agent.**   This is the setting we consider in the KeyQN experiments (sec. 4.2.1). However,
co-learning the representation (keypoints) with the agent is also important, especially for hard to explore environments.
This is because random exploration (currently used for pre-training Transporter) cannot cover all parts of the environment
(certain objects / rooms remain unseen). This is important future work.

**R4: Code release.**   Thank you, yes we will share our implementation of the Transporter model.

[Meta-Review · NeurIPS 2019]

The reviewers considered the self-supervision proxy problem interesting and novel and the experimental results convincing